# Analysis of Free Amino Acid Composition and Honey Plant Species in Seven Honey Species in China

**DOI:** 10.3390/foods13071065

**Published:** 2024-03-29

**Authors:** Jialin Yang, Yihui Liu, Zongyan Cui, Taohong Wang, Tong Liu, Gang Liu

**Affiliations:** 1College of Life Science, Shihezi University, Shihezi 832003, China; yjia8311161@163.com; 2Xinjiang Production and Construction Corps Key Laboratory of Oasis Town and Mountain-Basin System Ecology, Shihezi 832003, China; 3College of Bioscience and Biotechnology, Hunan Agricultural University, Changsha 410128, China; liuyihui@stu.hunau.edu.cn; 4Technology Center of Qinhuangdao Customs, Qinhuangdao 066004, China; ciqqhd@126.com (Z.C.); m17325381081@163.com (T.W.)

**Keywords:** honey, free amino acids (FAAs), UPLC-MS/MS, nectar plants

## Abstract

Honey is well-known as a food product that is rich in active ingredients and is very popular among consumers. Free amino acids (FAAs) are one of the important nutritional components of honey, which can be used not only as a nutritional indicator of honey but also as an indicator of plant source identification. In this study, the contents of 20 FAAs in seven types of honey from 11 provinces in China were examined for the first time. The 20 FAAs were analyzed by ultra-performance liquid chromatography-mass spectrometry/mass spectrometry (UPLC-MS/MS). By analyzing 93 honey samples from seven types of honey, the FAAs were found to range from 394.4 mg/kg (linden honey) to 1771.7 mg/kg (chaste honey). Proline ranged from 274.55 to 572.48 mg/kg, and methionine was only present in some of the linden honey, chaste honey, acacia honey, and rape honey. Evaluated by amino acid principal component analysis, multifloral grassland honey had the highest overall evaluation score, acacia and jujube honey were the most similar, while chaste honey was the least similar to the other types of honey. In addition, DNA was extracted from 174 Xinjiang grassland honey samples and different plant leaves for PCR and sequencing to identify the species of nectar plants. As a result, 12 families and 25 species of honey plants were identified. The results confirmed the diversity of FAAs in dissimilar types and sources of honey. This study provides a reference for expanding honey quality standards and verifying the authenticity of honey.

## 1. Introduction

Honey is a naturally sweet substance produced by bees after collecting nectar or secretions from living parts of plants and mixing them with their own secretions [1]. Honey is considered healthier than the consumption of sucrose and is not only used as a food source, but also has some medicinal properties [2,3]. Many studies have shown that honey has antioxidant [4], anti-inflammatory [5], and anti-obesity [6], anti-cancer [7], anti-atherosclerosis [8], neuroprotective [9], and immunomodulatory activity [10], etc.

Honey is a potential natural antioxidant, in which a variety of phenolic compounds play a major role. Dżugan et al. [11] studied the antioxidant activity of 90 kinds of Polish honey and found that the antioxidant activity was related to its color shade and phenolic content, with darker colors and higher phenolic content having stronger antioxidant activity. Honey has broad-spectrum antibacterial properties and can be antibacterial or antiseptic by direct or indirect action. Due to its antibacterial and anti-inflammatory effects, honey has a positive impact on wound healing. Honey accelerates tissue repair, reduces the risk of infection, and leads to faster wound healing [12]. In addition, honey is helpful in regulating the digestive system. It contains acids and enzymes that help promote intestinal health and relieve stomach upset, which is beneficial for relieving mild indigestion [13]. Wang et al. [14] demonstrated the protective effect of monofloral honey from *Prunella vulgaris* against dextran sodium sulfate-induced ulcerative colitis in rats by regulating gut microbiota.

The major components of honey include glucose, fructose, sucrose, water, proteins, and acids [15]. Of these, glucose and fructose are the capital sugar components in honey, making up the majority of the total sugar [16]. Honey contains a variety of enzymes, which are involved in the production process of honey, play a prominent role in the preservation and quality of honey, and are also capable of exerting certain biological activities in the human body [17]. In addition, honey has proteins, minerals, organic acids, and FAAs as well as various phytochemicals [18].

The physicochemical properties of honey are directly related to its source origin and place of origin, and honey quality is evaluated through standard quality analyses. Generally speaking, the water content of honey should be ≦20%, sucrose content should be ≦5%, 5-hydroxymethylfurfural content should be ≦40 mg kg^−1^, etc. [19]. The standard physicochemical index of honey from different nectar plants varies; for example, heather honey should have a moisture content ≦ of 23%, honeydew honey should have glucose and fructose ≧of 45%, and lavender honey should have a sucrose content ≦ of 15% [20].

Amino acid is the raw material of protein, it is the essential nutrient of honey plants, affecting their growth and development, and also one of the nutrients required by the human body [21]. FAAs are important nutritional components of honey, providing important information on the plant origin of honey and its quality, as well as providing some reference on whether the honey is adulterated or not [22]. Honey contains most types of amino acids, such as tyrosine, methionine, lysine, phenylalanine, histidine, glycine, etc. Since the nectar plant is the major provider of amino acids in honey, and geographic and floral sources can lead to subtle differences in amino acid types and content, some studies have indicated that differences in amino acids can be used as a means of identifying the floral source of the honey [23].

Various methods have been studied for the analytical determination of FAAs, and FAAs are usually determined by chromatography [24]. For the successful determination of FAAs, high-performance liquid chromatography with fluorescence detection (HPLC-FLD) and ultraviolet detection (HPLC-UV) have accumulated good results [25]. Azevedo et al. [26] proposed the determination of FAAs in Brazilian honey by GC-MS, but this method requires the use of solid-phase extraction followed by the addition of alkyl chloroformate reagents prior to analysis. However, the measures listed above require a derivatization step, making the whole determination process cumbersome and time-consuming. In addition, proton nuclear magnetic resonance (^1^H-NMR) has been used to detect FAAs in honey [27]. This method is fast and efficient but has the disadvantage of being more expensive and complex than conventional HPLC. Currently, UPLC-MS/MS has been widely used for the detection of amino acids due to its good sensitivity and selectivity.

Based on the literature review, there are few studies on the composition of FAAs in honey and their sources. Beneficial nutritional activities are significant to understand the value of honey and accurate, rapid, and easy analytical methods are needed to determine the FAAs composition of honey. In this study, a quick and effective method for the determination of 20 FAAs in various types of honey samples was developed based on UPLC-MS/MS detection. In the first phase, a method for the determination of FAAs in honey samples was proposed and validated. The second phase of the study aimed to determine the complete AA composition of honey from diverse plants and geographical sources. The third was to analyze the honey plant species of grassland multifloral honey with the highest amino acid scores. Different types of samples (n = 93) from Xinjiang, Shanxi, Shaanxi, Jilin, Heilongjiang, Liaoning, Anhui, Sichuan, Shanxi, Gansu, Qinghai, Henan, and Hebei were tested and chemometric analyses were performed to understand the great importance of the results obtained.

## 2. Materials and Methods

### 2.1. Samples

The material for the study was collected randomly over the three years of the study: 2021, 2022, and 2023. A total of 93 honey samples were analyzed: 6 monofloral honey samples (20 linden, 13 chaste, 19 locust, 11 rape, 5 jujube, and 2 lavandula) and 23 grassland honey samples. The honey samples were divided into two parts (200 g each, one for analysis and one for control) and stored in a refrigerator at 4–5 °C until the end of the analysis. Details including type of honey, number of samples, area of collection, date of collection, number of acquisitions, and beehive location are listed in Table 1.

### 2.2. Chemical Reagents

The analytical standards of 20 amino acids, including aspartic acid (Asp), glutamic acid (Glu), γ-am aminobutyric acid (GAMA), serine (Ser), glutamine (Gln), histidine (His), asparagine (Asn), glycine (Gly), threonine (Thr), arginine (Arg), alanine (Ala), tyrosine (Tyr), valine (Val), methionine (Met), tryptophan (Trp), phenylalanine (Phe), isoleucine (Ile), leucine (Leu), lysine (Lys), and proline (Pro), were obtained from Sigma-Aldrich Co. (St. Louis, Mo, USA) with all purity ≥ 99%. Phenyl isothiocyanate (PITC) (purity ≥ 99%) and triethylamine (purity ≥ 99%) were purchased from Sigma-Aldrich (St. Louis, MO, USA). Ammonium acetate acetonitrile (HPLC grade) and hexane (HPLC grade) were obtained from Dikma (Beijing, China). Ultra-pure water was prepared using a Milli-Q plus system (Millipore, Boston, Massachusetts, The United States of America).

The individual stock solutions were prepared in 0.1 mol/L HCl solution at the concentration of 10,000 μg/mL. The standard mixture was prepared from the individual stock standard solutions at the concentration of 100 μg/mL. The working solutions of the amino acid were prepared by serial dilutions. The internal standard norleucine was prepared with a concentration of 100 μg/mL. All the solutions were stored at 4 °C in the refrigerator.

### 2.3. Sample Preparation and Derivatization

The sample pretreatment procedure was performed as described by Ameri et al. [28] with some modifications. Briefly, an aliquot of honey sample (1.0 g) was weighed and transferred into a 50 mL centrifuge tube (Leibus Biological Co. Ltd., Shanghai, China), and 25 mL of 0.1 mol/L HCl solution was added. After mixing homogeneously on a vortex-genie2 (Scientific Industries, Austin, TX, USA) mixer for 5 min, 1 mL of the liquefied honey samples were transferred to a 10 mL centrifuge tube (Leibus Biological Co. Ltd., China), adding 0.5 mL of 1.2% phenyl isothiocyanate and 14% triethylamine, respectively. After mixing homogeneously, the derivatization reaction was carried out at room temperature for 1 h. Finally, 0.1 mL of 2% acetic acid and 2 mL n-hexane were added, vortexed for 2 min, and the solution was filtered through a 0.22 μm organic phase filter.

### 2.4. UPLC-MS/MS Analysis Conditions

Amino acid derivatives were analyzed using UPLC-MS/MS equipped with an ACQUITY UPLC BEH C18 (100 mm × 2.1 mm, 1.7 μm) column (Waters, Milford, MA, USA) at a column temperature of 40 °C and a flow rate of 0.3 mL/min. The mobile phase consisted of acetonitrile (A) and 10 mmol/L aqueous ammonium acetate (B). The linear gradient program was as follows: 1–2.5 min, 1% A; 2.5–5.5 min, 18% A; 5.5–6.5 min, 18–80% A; 6.5–7.5 min, 80% A; 7.5–8 min, 80–1% A; and 8–12.5 min, 1% A. The injection volume was 2 μL, and mass spectrometry was carried out in the positive mode with an electrospray ionization (ESI). Mass spectrometry (MS) analysis conditions were as follows: capillary voltage of 3.0 kV, lens voltage of 0.1 V, desolvation temperature of 350 °C, source temperature of 120 °C, desolvation gas (N_2_) flow rate of 650 L/h, cone gas (N_2_) flow rate of 50 L/h, and collision gas (Ar) flow rate of 0.15 mL/min.

### 2.5. Nucleic Acid Extraction and Quality Control in Grassland Honey

DNA extraction was performed as described by Utzeri et al. [29]. We took 5 μL DNA solution 1% agarose, 1X TAE buffer solution electrophoresis (voltage 120~180 V) detection; a single band indicates that the DNA is intact and has no degradation, and there are obvious bands indicating that the concentration can meet the PCR requirements. We used a spectrophotometer to detect the concentration and purity, took 1 μL to check the OD value, OD260/280 in 1.7~2.0, indicating that the DNA quality is good, with less than 1.7 protein contamination, and greater than 2.0 RNA contamination. Generally, there is a small amount of protein, and RNA contamination does not affect the ordinary PCR.

### 2.6. Design Primers

The primer design software adopts Primer Premier 5, and the setting parameters are as follows: primer length 18~30 bp; Tm value 55~65 degrees, annealing temperature about 60 degrees; GC content 40~70%. Special care was taken to avoid primer dimers and the presence of non-specific amplification, avoiding four consecutive bases, especially G and C, and preferably no more than three G or C within the last five bases at the 3′ end. The length of PCR amplification products was as follows: 150~300 bp for site sequencing primers, 80~150 bp from the site, and 150 bp upstream and downstream from the exon for exon detection primers; the PCR product of target gene sequencing is generally not more than 1200 bp. Three pairs of primer pairs were designed for amplification in this project, and the products with the best results were selected for subsequent sequencing analysis. ITS (Internal Transcribed Spacer) sequence analysis method has high reliability for plant phylogenetic species identification and evolutionary studies. The sequences of the primers in this study are shown in Table 2.

### 2.7. PCR Amplification

PCR amplification was conducted according to the method of Prosser et al. [30]. The PCR amplification system was as follows: 2× Taq Mix 10 μL, Forward Primer and Reverse Primer 1 μL each, template DNA 2 μL, made up to 20 μL with sterilized double-distilled water, and the amplification procedure was as follows: denaturation 95 °C for 5 min; annealing 95 °C for 30 s, 59 °C for 30 s, 72 °C for 1 min, 30 cycles; extension 72 °C for 10 min; PCR products were subjected to 1% agarose gel electrophoresis, and then detected in BIO-RGG gel electrophoresis. The PCR products were detected by 1% agarose gel electrophoresis, and the electrophoretic bands were detected in the BIO-RAD GelDoc XR+ gel imaging system.

### 2.8. Electrophoretic Detection and Recovery

We cut off the gel block containing the target fragment from the agarose gel and weighed it. We added buffer B2 3–6 times the weight of the gel block and solubilized in a 50 °C water bath for 5–10 min. We transferred the lysate to an adsorbent column and centrifuged at 10,000 rmp for 30 s. We poured off the liquid into the collection tube. We added 500 μL of Wash Solution, centrifuged at 10,000 rmp for 30 s, and poured off the liquid into the collection tube. We repeated the procedure once. We centrifuged the empty column at 10,000 rmp for 1 min. We placed the adsorbent column into a clean 1.5 mL centrifuge tube, added 15–40 μL of Elution Buffer to the center of the adsorbent membrane, and let stand at room temperature for 1 min before centrifuging for 1 min. We preserved the DNA solution in the tube.

### 2.9. Establishment of a Nectar Plant Bank

A total of 174 honey samples were collected from 30 apiaries within a radius of 1 km from June to August from flowering nectar plants in the multifloral grassland honey collection site in Ili, Xinjiang. At the same time, we collected flowering honey plants, extracted DNA from plant leaf samples (Rapid DNA Extraction Kit, TIANGEN, Beijing, China), performed PCR and sequencing for identification, and then analyzed the sequenced nucleotide sequences in basic local alignment search tool (BLAST) analysis on the NCBI website to identify the species of honey plants and establish a honey plant library.

### 2.10. Statistical Analyses

All results are presented as mean ± standard error. Analysis of variance (ANOVA) was used to compare sample means to analyze changes in observed parameters between honey samples. All data were statistically processed using the STATISTICA 12.0 software package. Principal component analysis (PCA) and cluster analysis (CA) were applied to the experimental data (used as descriptors) to characterize and differentiate the observed samples.

## 3. Results and Discussion

### 3.1. Method Validation

The optimum mass spectrometry parameters for all analytes are summarized in Table 3. Figure 1 shows the MRM mode chromatograms of the 20 amino acid derivatives. The linearity of the 20 amino acids was good in the concentration range of 1~500 mg/kg with correlation coefficients (R^2^) greater than 0.997. It was recommended that the limits of detection (LODs) and limits of quantification (LOQs) of the method be set at ≥10 and ≥3 S/N, respectively, and the LODs and LOQs of each analyte be set at 0.1 mg/kg and 0.3 mg/kg, respectively. The spiked recoveries ranged from 77.2% to 107.5% with the relative standard deviations (RSDs) of 0.6% to 9.7% at the spiked recoveries of 10 mg/kg and 100 mg/kg. The results showed that the method is suitable for the determination of FAAs in honey samples.

### 3.2. Determination of 20 FAAs in Honey

The mean values of FAAs depended on the type and origin of the honey (Table 4 and Figure 2). The highest total FAA concentration (mean concentration of 1771.7± mg/kg) was found in chaste honeys (n = 13). Among the other monofloral honeys, lavandula honeys (n = 2) contained FAA concentrations of 780.9± mg/kg and jujube honeys (n = 5) 719.3± 40 mg/kg. Total FAA concentrations were 476.1± mg/kg and 486.4± mg/kg in locust honeys (n = 19) and rape honey (n = 11), respectively, and, among monofloral honeys, linden honeys (n = 20) contained the lowest concentration of 394.4± mg/kg, and, in grassland honeys (n = 17), the total FAA concentration was 1061.9± mg/kg (Figure 2).

Proline (Pro) is the dominant FAA in honey, as it accounts for more than 50 percent of the total FAAs [31]. It is derived from the hemolymph of bees and nectar, while the other amino acids are derived from plant nectar, bees, and pollen. Pro has been suggested as one of the indicators of honey adulteration, and natural honey should contain more than 180 mg/kg of proline [32,33]. Pro content in honey can be used as a standard of honey maturity [34]. In this study, the proline content of several honeys was found to be the highest content of jujube honey, at 572.476 mg/kg, grassland honey, 566.035 mg/kg, chaste honey, 479.963 mg/kg, lavandula honey, 417.0 mg/kg, locust honey, 342.279 mg/kg, rape honey, 286.327 mg/kg, and linden honey, 274.548 mg/kg; Proline content was below 300 mg/kg in rape honey, at 286.327 mg/kg and linden honey, 274.548 mg/kg. If the Pro value of rape and linden honeys is low, it may be an indication of the low maturity of the honey. The reason for this may be that during the flowering period of linden and rape, there is more rainfall and cloudy days, the temperature is lower, the nectar collected from the honey is high in moisture content, and the honey is not yet fully matured at the time of sampling.

Glycine (Gly), as an excitatory neurotransmitter, contributes to the growth of the central nervous system in humans [35]. The mean concentration of Gly was determined in all types of honey samples: linden honey at 1.458 mg/kg, chaste honey, 3.701 mg/kg, locust honey, 1.645 mg/kg, rape honey, 2.254 mg/kg, jujube honey, 2.884 mg/kg, lavandula honey, 4.2 mg/kg, and grassland honey, 5.700 mg/kg.

Lysine (Lys), histidine (His), and arginine (Arg) are exogenous hexose alkaline amino acids, of which arginine is the most alkaline FAA of all proteins and plays a prominent role in metabolism. Rape honey samples contained the highest amount of arginine at 8.11 mg/kg. His was present in all the honey species, with the highest amount in the honey from grassland (mean concentration of 12.863 mg/kg), and the lowest amount in the honey from linden and jujube (4.547 mg/kg and 2.978 mg/kg, respectively). The honey samples were found to contain histidine, which is a basic amino acid that can be used as an antioxidant in the production of honey. Biluca et al. [22] did not find it in Brazilian honey samples from nine species. In contrast, His was dominant in honeydew honey from Malaysia (589–2777 mg/kg). Lys was highest in rape honey (33.472 mg/kg), grassland honey (33.111 mg/kg), and lavandula (22.6 mg/kg), while the rest of the honey contained between 5 and 18.2 mg/kg.

Methionine (Met) is a sulfur-containing essential AA and a precursor for cellular methylation and cysteine synthesis [36]. Met has a wide range of properties including chemical modification, cellular metabolism, and metabolic derivatives, and regulates the activity of many enzymes, and is essential for the metabolism of polyamines, creatine, and phosphatidylcholine [37]. Met is present at levels of 1.458–3.701 mg/kg in some linden, chaste, locust, and rape honeys, except for the grassland, lavandula, and jujube honeys. Sun et al. [38] did not find Met in five Chinese honey samples (rape, jujube, acacia, lungan, and chaste honey). Met was also not detected in multifloral honey samples from Madeira [39]. In addition, Iglesias et al. [40] studied 23 FAAs in honey by HPLC and found that methionine was absent from honey from the center of Spain.

Phenylalanine (Phe) is an exogenous aromatic AA responsible for transmitting signals between nerve cells in the central nervous system, as well as between TYR, one of the sugar-producing AAs. The average concentration of Phe in honey samples varies; we found 1049.337 mg/kg in chaste honey, 199.7 mg/kg in lavandula honey, 87.212 mg/kg in grassland honey, 39.205 mg/kg in linden honey, 14.696 mg/kg in jujube honey, 11.116 mg/kg in rape honey, and 5.781 mg/kg in locust honey. The phenylalanine content varies from one honey to another; for example, Brazilian and Serbian honey contained 1231 mg/kg and 80.8 mg/kg of Phe, respectively [41], whereas Phe was not detected in five different sources of honey from one region of France [42]. Karabagias et al. [43] noted a high variability in the Phe and Pro content of Greek honey from diverse geographical regions. The researchers’ observations suggest that the formation of Phe is likely to be related to a more favorable climate at lower latitudes.

Grassland honey contained an alarmingly high level of Tyr: 97.93 mg/kg. High tyrosine concentrations (mean 97.93 mg/kg) were only characteristic of the herbal honey samples from Ili, Xinjiang, China, whereas Tyr was reduced to 48 mg/kg and 42.1 mg/kg in the bramble and lavandula honey samples, respectively. The rest of the samples contained levels ranging from 3.39 to 8.23 mg/kg. The presence of Tyr can be considered an indicator of polyfloral grassland honey. Kuś [44] found 7.8–264 mg/kg of Tyr in seven honey species (buckwheat, black locust, goldenrod, canola, fir, and linden). In addition, Boffo et al. [45] found that wildflower honey has higher levels of phenylalanine and tyrosine.

Valine (Val), isoleucine (Ile), and leucine (Leu) are important branched-chain AAs [46]. They cannot be synthesized by the body, must be obtained from food, and have many beneficial effects, such as controlling blood glucose and alleviating skeletal muscle damage [47,48]. The highest total concentrations of Val + Ile + Leu were found in the honey of grassland (35.66 mg/kg) and chaste honey (20.58 mg/kg), and the lowest total concentration was found in locust honey (7.85 mg/kg) (Table 4). However, Janiszewska et al. [49] reported a higher total amount of branched-chain amino acids in buckwheat honey of 17.6 mg/100 g.

Total endogenous amino acids were highest in grassland honey (770 mg/kg) and lowest in linden honey (308.81 mg/kg), respectively. Total exogenous amino acids were highest in chaste honey (1159.18 mg/kg) and lowest in locust honey (44.95 mg/kg), respectively (Table 4). The highest exogenous/endogenous FAA ratio was found in jujube honey (up to 1.9) and the lowest exogenous/endogenous FAA ratio was found in jujube honey (0.08). This suggests that honey from grassland herbs in the Ili Steppe of Xinjiang is the most valuable for human consumption due to the relatively low level of environmental contamination in which grassland honey plants grow.

### 3.3. Chemometric Analysis

Figure 3A,B shows the effect of honey source on amino acid composition. Among the 20 amino acids from seven different sources of honey, Met and Ala had the highest negative correlation number (r = −0.82).The maximum values of Pearson correlation coefficients were similar to those of Thr and Ser (r = 1), Ile and Leu (r = 0.96), Leu and Val (r = 0.96), Leu and Gly (r = 0.93), Ala and Leu (r = 0.89), Val and Ile (r = 0.93), Ile and Gly (r = 0.89), Gly and Ala (r = 0.93), and Ala and Val (r = 0.86), suggesting that these amino acids are usually the most effective indicators of honey.

Based on the principal component eigenvalues greater than 1 and the cumulative contribution rate ≥80%, three principal components were extracted from the amino acids in the seven honeys, with a cumulative contribution rate of 80%, which can reflect most of the information on the content of the 20 FAAs in the seven honeys. The first principal component included Gly, Ala, GABA, Ser, Pro, Thr, Asp, Tyr, Ile, Leu, and Lys, the second principal component included Asp, Glutamine, His, Arg, Met, Phe, and Try, and the third principal component included Glu (Figure 3C). A principal component composite score model can be derived: F = 0.48F1 + 0.20F2 + 0.14F3, where F is the composite score of the honey, and F1, F2, and F3 are the scores of each principal component. The composite scores and rankings of the seven types of honey were calculated. When the bond distance is 0.087, the seven types of honey can be classified into three categories. The first type is linden, locust, jujube, rape and grassland honey; the second type is lavandula honey; and the third type is chaste honey, the results are shown in Figure 3D,E. Analyzed from the perspective of amino acid content, the composite rankings of the different types of honey, from highest to lowest, were as follows: grassland honey, chaste honey, jujube honey, lavandula honey, rape honey, locust honey, and linden honey. Based on the amino acid composition and data clustering, we conclude that locust and jujube honey are the most similar, reflected in the shortest distance, and are similar in composition to linden honey, whereas chaste honey is the least similar to the other types of honey, suggesting that it has a unique amino acid composition.

The experiment extracted DNA from the leaves of 53 nectar plants, performed PCR and sequencing for identification, and a total of 40 samples were successfully identified, with 34 plants and one fungus identified, which were distributed in 15 families and 34 genera/species. Among them, *Asteraceae*, *Labiatae*, and *Leguminosae* were the top three, with 10, 5, and 3 genera/species, respectively. The DNA extracted from the honey from the multifloral grassland in Ili, Xinjiang was compared with a honey plant library, and a total of 12 families and 25 species of plants were identified from the DNA sequencing results of the plant source in honey; they were, in descending order of occurrence, were *Asteraceae* (six species), *Labiatae* (five species), *Leguminosae* (three species), *Campanulaceae* (two species), *Rosaceae* (two species), *Umbelliferae* (one species), *Zygophyllaceae* (one species), *Cruciferae* (one species), *Solanaceae* (one species), *Gentianaceae* (one species), *Orobanchaceae* (one species), and *Ranunculaceae* (one species) (Figure 3F).

The most detected nectar plants in the 174 samples were *Sisymbrium loeselii* (22.4%) and *Trifolium pratense* (14.94%), and the other nectar plants, in descending order of abundance, were *Thymus vulgaris*, *Ballota nigra*, *Salvia deserta*, *Prunus triloba*, *Datura stramonium*, *Lathyrus pratensis*, *Taraxacum erythrospermum*, *Campanula glomerata*, *Codonopsis clematidea*, *Lavandula angustifolia*, *Rhinanthus minor*, *Echinops tjanschanicus*, *Heracleum candicans*, *Origanum vulgare*, *Gentianella turkestanorum*, *Sophora moorcroftiana*, *Aconitum tanguticum*, *Agrimonia pilosa*, *Silybum marianum*, *Arctium tomentosum*, *Carthamus tinctorius*, *Zygophyllum macropodum* and *Onopordum acanthium* (Figure 3G).

## 4. Conclusions

A method for the determination of 20 FAAs in honey was established and applied. The results of method validation showed that the method is suitable for the determination of FAAs in honey samples and that the UPLC-MS/MS technique is reliable for the detection and identification of large amounts of FAAs in dissimilar kinds of honey. In this study, it was found that jujube honey had the highest content of proline compared to several other kinds of honey. Gly was determined in all types of honey samples with small differences in content. Arg was highest in rape honey. His, Tyr, branched-chain amino acids, and endogenous AAs were the most abundant in multifloral grassland honey. Met was present in linden honey, chaste honey, acacia honey, and rape honey. Phe was found to be the most abundant in chaste honey. In addition, it was shown that the content of individual AAs and the ratio of exogenous/endogenous FAAs varied depending on the type of honey and the place of origin, which can be used to practically verify the authenticity of honey. According to chemometric analysis, honey from the grassland in Xinjiang had the highest combined amino acid score, while acacia and jujube honey had the highest similarity. The results of this study are informative for the detection of FAAs in honey and verifying the authenticity of honey.

## Figures and Tables

**Figure 1 foods-13-01065-f001:**
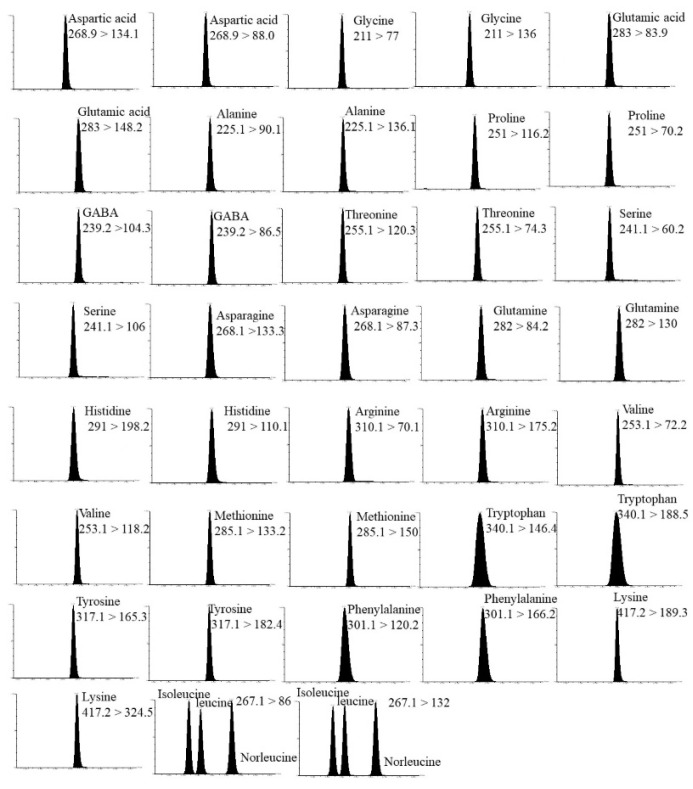
MRM chromatograms of 20 amino acid derivatives and internal standards (removal of methylleucine).

**Figure 2 foods-13-01065-f002:**
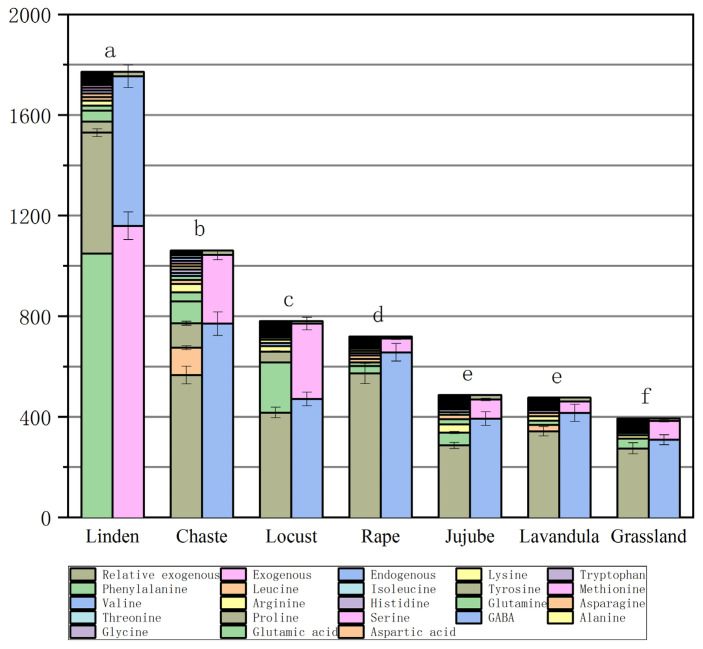
Figure of total free amino acid concentrations (column 1) and total endogenous, exogenous, and relative exogenous amino acid concentrations (column 2) in different types of honey. (The same letter above the bar graph indicates that the value is not statistically significant (*p* ≥ 0.05)).

**Figure 3 foods-13-01065-f003:**
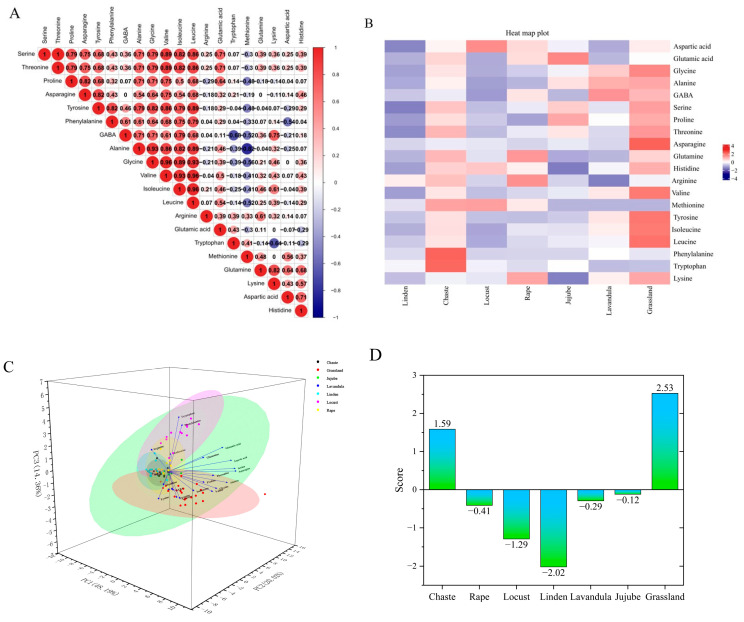
(**A**) Heat map of Pearson’s correlation coefficient relationship between FAAs in all studied honey types. (**B**) Heat map of FAA content in different types and sources of honey. (**C**) Three-dimensional plot of principal component analysis of seven honey species. (**D**) Figure of composite scores for different types of honey. (**E**) Figure of cluster analysis of different types of honey. (**F**) Figure of the self-built nectar plant bank and the nectar plants detected from the honey samples (blue). (**G**) Figure of the classification and number identified in multifloral grassland honey.

**Table 1 foods-13-01065-t001:** Table of details of honey samples collected.

Type of Honey	Number of Samples	Area of Collection	Date of Collection	Number of Acquisitions	Beehive Location
Linden honey	L1-L14	Jilin	July 2021	20	Yanbian Prefecture, Jilin Province, China, mountainous area, 128°54′ N, 43°7′ E
	L15-L16	Jilin	July 2022		
	L17	Heilongjiang	July 2023		China, mountainous area of Raohe County, Heilongjiang, 134°01′ N, 46°48′ E
	L18	Heilongjiang	July 2021		
	L19-L20	Heilongjiang	July 2022		
Chaste honey	T21	Shanxi	June 2022	13	China, Shanxi Lingshi County area, 111°47′ N, 36°51′ E
	T22, T23	Shanxi	June 2021		
	T24-T25	Liaoning	June 2023		China, Liaoning Chaoyang County area, 120°23′ N, 41°29′ E
	T26-T28	Hebei	June 2023		China, Hebei Qianxi County area, 118°19′ N, 40°8′ E
	T29	Hebei	June 2022		
	T30	Hebei	June 2023		
	T31	Hebei	June 2022		
	T32	Henan	June 2022		China, Yichuan County area, Henan, 112°25′ N, 34°25′ E
	T33	Henan	June 2021		
Locust honey	LO34	Gansu	June 2023	19	China, Tianshui County area, Gansu, 105°53′ N, 34°34′ E
	LO35-LO36	Gansu	May 2022		
	LO37-LO39	Hebei	May 2023		China, Hebei Qianxi County area, 118°19′ N, 40°8′ E
	LO40-LO41	Shanxi	May 2023		China, Shanxi Lingshi County area, 111°47′ N, 36°51′ E
	LO42	Shanxi	June 2021		
	LO43-LO48	Shaanxi	June 2023		China, Yichuan County area, Yan’an City, Shaanxi Province, 110°8′ N, 36°17′ E
	LO49-LO52	Shaanxi	June 2022		
Rape honey	RA53-RA55	Anhui	March 2022	11	China, Anhui Susong County area, 116°56′ N, 30°9′ E, Agricultural land
	RA56-RA57	Sichuan	March 2022		China, Cangxi County Area, Guangyuan City, Sichuan, 105°56′ N, 31°44′ E, Agricultural Land
	RA58-RA60	Sichuan	March 2023		
	RA61-RA63	Qinghai	July 2023		China, Qinghai Menyuan County Area, 101°37′ N, 37°22′ E, Agricultural Land
Jujube honey	JU64	Shanxi	June 2021	5	China, Shanxi Region, Mountainous Area, 110°59′ N, 37°57′ E, Agricultural Land
	JU65	Shanxi	June 2021		
	JU66	Shanxi	May 2022		
	JU67	Shanxi	May 2023		
	JU68	Shanxi	June 2022		
Lavandula honey	LA69-LA70	Xinjiang	July 2023	2	China, Yili Prefecture, Huocheng County Area, 83°25′ N, 43°44′ E, Wild Natural Grassland
Grassland honey	GR71-GR93	Xinjiang	July 2023	23	China, Nilek County District, Yili Prefecture, Mountainous Area, 80°54′ N, 44°17′ E, Agricultural Land

**Table 2 foods-13-01065-t002:** Table of ITS primer sequences.

Primer Name	Sequences
ITS1-F	5′-TCCGTAGGTGAACCTGCGG-3′
ITS1-R	5′-TCCTCCGCTTATTGATATGC-3′
ITS2-F	5′-ATGCGATACTTGGTGTGAAT-3′
ITS2-R	5′-GACGCTTCTCCAGACTACAAT-3′
ITS3-F	5′-GGAAGTAAAAGTCGTAACAAGG-3′
ITS3-R	5′-TCCTCCGCTTATTGATATGC-3′

**Table 3 foods-13-01065-t003:** Table of MRM parameters for 20 amino acid derivatives and internal standard (removal of methylleucine).

Amino Acid Types	Quantitative Ion Pair/(m/z)	Qualitative Ion Pairs/(m/z)	Monitoring Time/(s)	Foraminal Voltage/(V)	Crash Voltage/(V)
Aspartic acid	268.9/134.1	268.9/88	0.1, 0.1	40, 40	10, 20
Glutamic acid	283/148.2	283/83.9	0.1, 0.1	40, 40	10, 30
Glycine	211/136	211/77	0.02, 0.02	30, 30	15, 30
Alanine	225.1/90.1	225.1/136.1	0.02, 0.02	35, 35	10, 20
GABA	239.5/86.5	239.5/104.3	0.02, 0.02	35, 35	15, 10
Serine	241.1/106	241.1/60.2	0.02, 0.02	35, 35	10, 20
Proline	251/116.2	251/70.2	0.02, 0.02	40, 40	15, 25
Threonine	255.1/120.3	255.1/74.3	0.02, 0.02	35, 35	10, 20
Asparagine	268.1/133.1	268.1/87.3	0.02, 0.02	35, 35	10, 20
Glutamine	282/84.2	282/130	0.02, 0.02	40, 40	25, 20
Histidine	291/198.2	291/110.1	0.02, 0.02	35, 35	10, 25
Arginine	310.1/175.2	310.1/70.1	0.02, 0.02	40, 40	35, 15
Valine	253.1/72.2	253.1/118.2	0.05, 0.05	35, 35	15, 10
Methionine	285.1/150	285.1/133.2	0.05, 0.05	30, 30	10, 20
Tyrosine	317.1/165.3	317.1/182.4	0.05, 0.05	35, 35	15, 10
Phenylalanine	301.1/166.2	301.1/120.2	0.05, 0.05	40, 40	10, 20
Tryptophan	340.1/188.5	340.1/146.4	0.05, 0.05	40, 40	15, 30
Lysine	417.2/189.3	417.2/324.5	0.05, 0.05	35, 35	20, 15
Leucine	267.1/86	267.1/132	0.1, 0.1	40, 40	20, 10
Isoleucine	267.1/86	267.1/132	0.1, 0.1	40, 40	20, 10
Norleucine(IS)	267.1/86	267.1/132	0.1, 0.1	40, 40	20, 10

**Table 4 foods-13-01065-t004:** Table of free amino acid content determined by LC-MS/MS for different types of honey.

	Sum of Endogenous FAAs (mg/kg)	Sum of Exogenous FAAs (mg/kg)	Sum of FAAs (mg/kg)	SD	Ratio of Exogenous/Endogenous FAAS	Sum of Hydrophobic Pyruvate Derivates (mg/kg) Val + Ile + Leu	Max Concentrations of FAAs (mg/kg)	Min Concentrations of FAAs (mg/kg)	Median Concentrations of FAAs (mg/kg)
Linden	308.81	75.38	394.4	98.35	0.24	8.10	670.370	244.800	354.136
Chaste	594.57	1159.18	1771.7	116.02	1.95	20.58	2282.900	1295.270	1784.832
Locust	415.34	44.95	476.1	39.92	0.11	7.85	652.500	328.190	476.570
Rape	393.03	75.68	486.4	54.14	0.19	12.68	617.800	350.540	461.840
Jujube	656.53	55.49	719.3	119.90	0.08	13.94	847.160	599.052	680.223
Lavandula	470.89	299.40	780.9	340.93	0.64	18.05	933.308	628.580	780.944
Grassland	770.31	273.75	1061.9	69.14	0.36	35.66	670.370	244.800	354.136

## Data Availability

The original contributions presented in the study are included in the article, further inquiries can be directed to the corresponding authors.

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
