# Peer review of "Analysis of Free Amino Acid Composition and Honey Plant Species in Seven Honey Species in China"

_foods, 2024, doi:10.3390/foods13071065_

Round 1
Reviewer 1 Report
Comments and Suggestions for Authors
Dear authors,
The manuscript entitled “Analysis of free amino acids composition and honey plant species of seven honey species in China” is an interesting and well-structured study that expands knowledge on characterization and authentication of honeys based on their analysis of free amino acids composition. Also, it has an adequate literacy basis. However, it needs some minor revision to improve its understanding, following these suggestions:
-Line 12: Authors set “Honey is a food with a unique flavor and high nutritional value”….However, it is not true due to honey is mainly sugars (80-85%) and water (15-20%). Its real value is based on its beneficial properties (antioxidants, enzymatic complex and so on). Then, the sentence should be modified in this sense.
-Line 60: Please, check cite. It shows name initials. Same thing in line 286.
-Lines 65-69: In this paragraph authors refer to quality standard for honey but only shows exceptional criterion for humidity of 23 %. The regular criterion is £ 20%. Also, sucrose criterion in general is £ 5%. Please, introduce references for international standard (Codex) or European quality standard.
-Line 90: What does mean the number 1 superscript in the acronym?
-Lines 136-137: It is cited Kong et al. but the reference is different (Xiang-Hong et al., 2010). Please, check it.
-Line 179: Table 2 is not referenced in text, neither the acronym ITS.
- Line 184: Please, check the sentence. There is a dot between water and amplification.
-Line 206: What is BLAST analysis? Please, develop the acronym.
-Line 227: Table 3 shows parameters for 20 amino acids. However, they seem to be 21 amino acids. Same thing in figure 1.
-Line 241: Table 4 should improve its appearance. It has bad correspondence of rows for Twig chaste.
-Line 243: In figure 2 it would be Linden, not Liden.
-Line 259-260: Please, add better explanation for this lack of maturity in rape and linden honeys.
-Line 271: “2.97754 mg/kg”. This result shows 5 decimals, is there any reason?
-Line 306: After the dot it must be capital letter.
- Line 381: What does mean 3F?
- Line 388: The term respectively has not any sense in this context. Please, rewrite.
Author Response
Dear reviewer,
Thank you for your careful and kind feedback on our manuscript. We have revised the manuscript based on your comments and have carefully proofread the manuscript to minimize typographical, grammatical, and referencing errors. The changes made in response to the reviewers' comments are detailed below.
Comment from the reviewer:
Comment1:
-Line 12: Authors set “Honey is a food with a unique flavor and high nutritional value”….However, it is not true due to honey is mainly sugars (80-85%) and water (15-20%). Its real value is based on its beneficial properties (antioxidants, enzymatic complex and so on). Then, the sentence should be modified in this sense.
Response:
Thank you for your suggestion, in lines 12-13 we have changed the sentence to "Honey is well-known as a food product that is rich in active ingredients and is very popular among consumers. "
Comment2:
-Line 60: Please, check cite. It shows name initials. Same thing in line 286.
Response:
We have changed the format of the author's name in line 61 and line 296.
Comment3:
-Lines 65-69: In this paragraph authors refer to quality standard for honey but only shows exceptional criterion for humidity of 23 %. The regular criterion is £ 20%. Also, sucrose criterion in general is £ 5%. Please, introduce references for international standard (Codex) or European quality standard.
Response:
We have added the requirements of some physical and chemical indicators for honey in general and the references. See lines 67-69, "Generally speaking, the water content of honey should be ≦20%, sucrose content should be ≦5%, 5-hydroxymethylfurfural content should be ≦40 mg kg-1, etc. (Seraglio et al., 2019)."
Comment4:
-Line 90: What does mean the number 1 superscript in the acronym?
Response:
1H-NMR also refers to proton nuclear magnetic resonance, usually the preceding 1 needs to be superscripted, and there are many examples in the literature, refer to the following:
't Hart BA, Vogels JT, Spijksma G, Brok HP, Polman C, van der Greef J. 1H-NMR spectroscopy combined with pattern recognition analysis reveals characteristic chemical patterns in urines of MS patients and non-human primates with MS-like disease. J Neurol Sci. 2003 Aug 15;212(1-2):21-30. doi: 10.1016/s0022-510x(03)00080-7. PMID: 12809995.
Aswathi KN, Shirke A, Praveen A, Chaudhari SR, Murthy PS. Pulped natural/honey robusta coffee fermentation metabolites, physico-chemical and sensory profiles. Food Chem. 2023 Dec 15;429:136897. doi: 10.1016/j.foodchem.2023.136897. Epub 2023 Jul 18. PMID: 37480775.
Comment5:
-Lines 136-137: It is cited Kong et al. but the reference is different (Xiang-Hong et al., 2010). Please, check it.
Response:
Thank you for pointing this out, we are very sorry for this problem. After checking, we found that the DOI number of this article is wrong, this article is a Chinese paper, and the original DOI number cannot be searched on the website, so in order to be a more standardized citation, we have cited another English reference. Please see line 140 for details.
Ameri, M., & Daryanavard, S. M. (2023). Experimental Design Application for Measuring Histamine in Tuna Fish Samples by Phenyl Isothiocyanate Derivation Method Using Ultra-Performance Liquid Chromatography. J Chromatogr Sci, bmad060. doi:10.1093/chromsci/bmad060
Comment6:
-Line 179: Table 2 is not referenced in text, neither the acronym ITS.
Response:
We have already cited ITS and Table 2 in chapter 2.6, in lines 182-185: "ITS (Internal Transcribed Spacer) sequence analysis method has high reliability for plant phylogenetic species identification and evolutionary studies. the sequences of the primers in this study are shown in Table 2.".
Comment7:
- Line 184: Please, check the sentence. There is a dot between water and amplification.
Response:
In line 191, we checked and modified this sentence to read: "double-distilled water, and the amplification procedure was as follows".
Comment8:
-Line 206: What is BLAST analysis? Please, develop the acronym.
Response:
BLAST refers to the basic local alignment search tool, to which we give the full name on line 214.
Comment9:
-Line 227: Table 3 shows parameters for 20 amino acids. However, they seem to be 21 amino acids. Same thing in figure 1.
Response:
Thank you for your question. Norleucine in Table 3 and Figure1 is internal standard (IS), not measured amino acid. Its function is to eliminate the influence of the instrument during the quantitative analysis, and play the role of calibration, so that the results will be more accurate.
Comment10:
-Line 241: Table 4 should improve its appearance. It has bad correspondence of rows for Twig chaste.
Response:
We have reformatted Table 4 so that there is no other extra spacing.
Comment11:
-Line 243: In figure 2 it would be Linden, not Liden.
Response:
Thank you for your discovery. We have changed "Liden" to "Linden" in figure 2, and we have checked all the figures in the text and found that in some of them the word was also misspelled, which we have corrected.
Comment12:
-Line 259-260: Please, add better explanation for this lack of maturity in rape and linden honeys.
Response:
We have added possible reasons for the low maturity of rape and linden honey in the article, see lines 267-270: "The reason for this may be that during the flowering period of linden and rape, there is more rainfall and cloudy days, the temperature is lower, the nectar collected from the honey is high in moisture content and the honey is not yet fully matured at the time of sampling."
Comment13:
-Line 271: “2.97754 mg/kg”. This result shows 5 decimals, is there any reason?
Response:
Thank you for pointing out the problem, we have changed the five decimal places in line 281 to three to read "2.978".
Comment14:
-Line 306: After the dot it must be capital letter.
Response:
We have capitalized the word in line 316 to read "The".
Comment15:
- Line 381: What does mean 3F?
Response:
The 3G plot (Figure 3F has been changed to 3G due to the addition of an additional figure) refers to the types of nectar plants identified from Xinjiang multifloral grassland honey. It indicates the number of samples detected for each nectar plant (n=25) as a percentage of the total number of samples (n=174), so that we can see the percentage of nectar plants occupied in the steppe honey.
Comment16:
- Line 388: The term respectively has not any sense in this context. Please, rewrite.
Response:
In line 405, we have deleted "respectively".
According to the comments from all referees, we have revised the entire manuscript. Overall, we believe that the current version of the manuscript has been greatly improved. We would like to express our gratitude to the reviewers for their input. We hope that the revised version meets the requirements, and we look forward to receiving your feedback.
Yours sincerely.

Reviewer 2 Report
Comments and Suggestions for Authors
This manuscript is dedicated to the study of free amino acids in 93 Chinese honey samples from different regions; in addition, DNA was extracted from the leaves of honey plants from the same regions, and PCR and sequencing for identification was performed. Chemometric analysis was applied, the results are informative for the detection of FAAs in honey and verifying the authenticity of honey. The results are of interest, however there are several points which need additional attention by the Authors;
1. Abstract: The DNA studies have to be mentioned in the Abstract., and the study of nectar plants explained.
2. Abstract: what is meant by “categories” of honey? Please explain.
3. Abstract: 93 samples and 174 samples ate mentioned. What is the actual number?
4. “Twig chaste” – do you mean chaste tree?
5. Page 5, lines 136 – 137: There is a problem with this reference: The DOI corresponds to a completely different article. The cited article cannot be accessed because, on the website of the Journal, there is no issue 4 of volume 29 (2010). Moreover, no Kong in the references list. Please clarify.
6. Page 12, the first paragraph: Please present the PCS 3D plot.
7. Page 13, lines 395-396: …” honey from the Ili 395 grassland in Xinjiang was the most valuable for health …”. Based on what?
Author Response
Dear reviewer,
Thank you for your careful and kind feedback on our manuscript. We have revised the manuscript based on your comments and have carefully proofread the manuscript to minimize typographical, grammatical, and referencing errors. The changes made in response to the reviewers' comments are detailed below.
Comment from the reviewer:
Comment1:
Abstract: The DNA studies have to be mentioned in the Abstract., and the study of nectar plants explained.
Response:
Thank you for your suggestion. We have written about the study of DNA and honey plants in the abstract, as detailed in lines 23-25: "In addition, DNA was extracted from 174 Xinjiang grassland honey samples and different plant leaves for PCR and sequencing to identify the species of nectar plants. As a result, 12 families and 25 species of honey plants were identified. "
Comment2:
Abstract: what is meant by “categories” of honey? Please explain.
Response:
For a more prescriptive presentation, we have replaced "Categories" with "types" in line 18.
Comment3:
Abstract: 93 samples and 174 samples ate mentioned. What is the actual number?
Response:
Ninety-three samples (seven types of honey) were determined for amino acids, and 174 samples (grassland honey, of which both amino acids and nectar plant species were determined in 23 samples) were determined for nectar plants.
Comment4:
“Twig chaste” – do you mean chaste tree?
Response:
Thank you for your suggestion, we have changed "twig chaste" to "chaste" in the text for the sake of a more standardized presentation.
Comment5:
Page 5, lines 136 – 137: There is a problem with this reference: The DOI corresponds to a completely different article. The cited article cannot be accessed because, on the website of the Journal, there is no issue 4 of volume 29 (2010). Moreover, no Kong in the references list. Please clarify.
Response:
Thank you for pointing this out, we are very sorry for this problem. After checking, we found that the DOI number of this article is wrong, this article is a Chinese paper, and the original DOI number cannot be searched on the website, so in order to be a more standardized citation, we have cited another English reference. Please see line 140 for details.
Ameri, M., & Daryanavard, S. M. (2023). Experimental Design Application for Measuring Histamine in Tuna Fish Samples by Phenyl Isothiocyanate Derivation Method Using Ultra-Performance Liquid Chromatography. J Chromatogr Sci, bmad060. doi:10.1093/chromsci/bmad060
Comment6:
Page 12, the first paragraph: Please present the PCS 3D plot.
Response:
Thank you for your suggestion, I wonder if by "PCS" you mean "PCA"? We added and showed a 3D image of PCA in the text. We have changed this image to Figure 3C.
Comment7:
Page 13, lines 395-396: …” honey from the Ili 395 grassland in Xinjiang was the most valuable for health …”. Based on what?
Response:
Thank you for pointing out that we have misrepresented this sentence. In lines 412-413, we have changed the sentence to read: "honey from the grassland in Xinjiang had the highest combined amino acid score".
According to the comments from all referees, we have revised the entire manuscript. Overall, we believe that the current version of the manuscript has been greatly improved. We would like to express our gratitude to the reviewers for their input. We hope that the revised version meets the requirements, and we look forward to receiving your feedback.
Yours sincerely.
